# Severe Pneumonia and Sepsis Caused by *Dialister pneumosintes* in an Adolescent

**DOI:** 10.3390/pathogens10060733

**Published:** 2021-06-10

**Authors:** Maximilian Kaiser, Meike Weis, Katharina Kehr, Verena Varnholt, Horst Schroten, Tobias Tenenbaum

**Affiliations:** 1Pediatric Infectious Diseases, University Children’s Hospital Mannheim, Heidelberg University, 68167 Mannheim, Germany; Maximilian.Kaiser@umm.de (M.K.); Horst.Schroten@umm.de (H.S.); 2Pediatric Surgery, University Children’s Hospital Mannheim, Heidelberg University, 68167 Mannheim, Germany; 3Clinic of Radiology and Nuclear Medicine, Medical Faculty Mannheim, Heidelberg University, 68167 Mannheim, Germany; Meike.Weis@umm.de; 4Institute for Medical Microbiology and Hygiene, Medical Faculty Mannheim, Heidelberg University, 68167 Mannheim, Germany; Katharina.Kehr@umm.de; 5Clinic of Neonatology, Medical Faculty Mannheim, Heidelberg University, 68167 Mannheim, Germany; Verena.Varnholt@umm.de

**Keywords:** *Dialister pneumosintes*, bacteremia, sepsis, pneumonia, children

## Abstract

Background: *Dialister pneumosintes* (*D. pneumosintes*) is known to cause dental, periodontal or sinus infections. To date, the pathogen has only been described in a small number of cases with a severe infection. Case report: We describe the clinical case of a 13-year-old, obese female patient that presented with acute respiratory failure and sepsis. A CT-scan showed extensive bilateral patchy areas, subpleural and peribronchovascular consolidations with surrounding ground-glass opacity, extensive consolidations in the lower lobes of both lungs matching to a severe pneumonia and clinically emerging acute respiratory distress syndrome. Moreover, it showed extensive sinusitis of the right sinus frontalis, maxillaris and right cellulae ethmoidales. *D. pneumosintes* was isolated from an anaerobic blood culture obtained at admission. The antibiotic treatment included piperacillin/tazobactam and oral switch to ampicillin/sulbactam plus ciprofloxacin. Conclusions: We describe the first adolescent with severe systemic *D. pneumosintes* infection. Since the pathogen is difficult to culture the systemic virulence remains unclear. This work aims to sensitize health care specialists to consider *D. pneumosintes* infection in patients with periodontal or sinusal infection.

## 1. Introduction

*Dialister pneumosintes* (*D. pneumosintes*) is a small, non-fermentative, obligatory anaerobic, gram-negative rod-shaped bacillus [1]. It was first isolated from a nasopharyngeal sample during the flu epidemic in the early 20th century by Olitsky and Gates [2]. It has been frequently reported to cause gingivitis or periodontitis in up to 83% [1,3]. So far only a few cases of systemic infection have been described [4,5]. In this case report we describe the clinical course of a 13-year-old, female, obese patient with severe pneumonia leading to a sepsis and acute respiratory distress syndrome (ARDS) due to a systemic infection with *D. pneumosintes*. Moreover, we performed a review of the literature analyzing systemic cases. 

## 2. Case Report and Review of Literature

A 13-year-old female obese patient presented in the pediatric emergency department with symptoms of an acute sepsis with high fever (40.8 °C), impaired consciousness (Glasgow Coma Scale of 6), low blood-pressure (not measurable), tachycardia of 188 bpm, cyanosis and symptoms of a respiratory tract infection. She had slight flu-like symptoms already for 3 weeks that have worsened in the past 3 days.

The patient was immediately stabilized with crystalloid volume and admitted to the pediatric intensive care unit for suspected sepsis of unknown origin. Shortly after admission she needed intensified volume management and continuous catecholamine administration for further circulatory stabilization. Initial pulmonary support was performed with a high-flow-nasal-cannula (HFNC) therapy. As the pulmonary condition worsened further, she was intubated and mechanical ventilated within 24 h after admission. Tidal volumes were continuously adapted during ventilation for acute lung injury treatment.

Laboratory tests showed a high C-reactive protein (CRP) of 20.2 mg/dL and interleukin-6 of 4300 pg/mL along with a compromised renal function with urea of 65 mg/dL and slightly elevated liver enzymes (ASAT 75 U/L, gGT 76 U/L).

A CT scan showed extensive bilateral patchy areas, subpleural and peribronchovascular consolidations with surrounding ground-glass opacity, extensive consolidations in the lower lobes of both lungs matching to a severe pneumonia and clinically emerging ARDS. Moreover, it showed an extensive sinusitis of the right sinus frontalis, maxillaris and right cellulae ethmoidales. The images of Figure 1 show the radiological evidence in the performed chest X-rays, Figure 2 the sequential CT imaging.

The cause for the pulmonary infection and sepsis was *D. pneumosintes*, recovered from one venous blood sample in BD BACTEC™—PLUS anaerob/F-vial using BD BACTEC™ automated blood culture system (Becton, Dickinson, Franklin Lakes, NJ, USA). When identified positive after a time-to-positivity (TTP) of 34 h the sample was subcultured on various solid culture media. Growth was observed only on Schaedler-agar (Becton, Dickinson, Franklin Lakes, NJ, USA) after 48 h of incubation. The Gram-staining showed gram-negative rods which were identified as *D. pneumosintes* by MALDI-TOF (Microflex LRF, Bruker Corporation, Billerica, MA, USA). Minimum inhibitory concentrations (MICs) could not be determined due to the lack of growth when subcultivated on Brucella blood agar (Mast Diagnostica GmbH, Reinfeld, Germany). Furthermore, *Staphylococcus aureus* was further isolated in a tracheal secretion sample as a sign of bacterial co-pathogen superinfection. Repetitive SARS-Cov-2 testing was negative.

With persistent clinical critical condition and elevated levels of CRP (19.1 mg/dL), procalcitonin (7.35 μg/L) and leukocytes (22.4 Gpt/L) a whole-body CT scan was initiated for reevaluation 5 days after admission. It showed bipulmonary diffuse disseminated abscess-forming round foci with an increasing consolidation and abscess formation in comparison to the initial CT scan at admission (Figure 2).

Initial treatment included ceftriaxon for potential meningitis, but antibiotic treatment was switched to piperacillin/tazobactam after an unsuspicious lumbar puncture (culture as well as Biofire Filmarray^TM^, Salt Lake City, UT, USA). After 8 days of antibiotic treatment and clinical stabilization we were able to extubate the patient permanently. Ten days after admission the patient developed an acute itching maculopapular rash. We therefore switched the antibiotic treatment again to meropenem and added a specific anti-inflammatory treatment with systemic steroids and antihistamines.

The clinical condition as well as the laboratory tests showed a slow but steady improvement, so we could dismiss the patient from clinical treatment after a total of 17 days of intravenous antibiotic therapy. Oral antibiotic treatment (switched to ciprofloxacin and sultamicillin) was continued for two more weeks and discontinued at an out-patient visit with CRP and leukocyte levels within a normal range. The CT on day 65 after initial presentation showed almost complete resolution of the radiological findings.

We further reviewed the existing literature on systemic *D. pneumosintes* infections through a systematic literature review in “Pubmed” searching for “Dialister” AND “pneumonia” AND “bacteremia” AND “sepsis” AND “abscess”. The literature review revealed that *D. pneumosintes* has been described to cause systemic infections such as bacteremia in 6 case reports. Among them one patient died due to the complications of the infection. In 4 cases end-organ abscesses in the brain (2 cases), liver and mediastinum are presented. Concerning a pulmonary focus there were 2 cases described, one in a trauma patient with ventilation associated pneumonia (VAP) (Table A1).

## 3. Discussion

This is the first described case of a severe systemic infection of *Dialister pneumosintes* in an adolescent with severe pneumonia. The origin of the systemic spread can most likely be found in the extensive sinusitis of the right sinus frontalis, maxillaris and right cellulae ethmoidales diagnosed in the CT scan at hospital admission.

As Kogure and colleagues described previously, there is an apparent risk of a systemic bacteremia caused by a severe sinusitis [6], even though we did not cultivate *D. pneumosintes* from a sinusal fluid sample. Since *D. pneumosintes* is mainly found in the nasopharyngeal cavity, there is a high possibility that the sinusitis originates from a naso-buccal translocation as seen previously [7,8]. Moore et al., could show that *D. pneumosintes* constitutes part of the oral bacterial flora, especially in subgingival sites and can cause gingivitis and periodontitis in children and young adults [9].

Our patient became severely ill, needed prolonged mechanical ventilation and intensive care treatment, despite lack of risk factors in the personal history apart of being obese. Castellanos and colleagues described a severe pneumonia due to an infection of *D. pneumosintes* in a patient with COPD [10] and Bahrani-Mougeot and colleagues identified *D. pneumosintes* among other pathogens in a trauma patient with ventilation-associated pneumonia [11]. Overall, our clinical case shows that an immunocompetent patient with no personal history of pulmonary disease can suffer from severe clinical courses due to *D. pneumosintes*. Furthermore, patients with an extended history of risk factors can present severe clinical courses, as described in one case leading to death [5]. Risk factors such as obesity may contribute the disease severity and respiratory failure in our case as well [12].

Interestingly, in our case we observed multiple small abscess formations in the course of pneumonia, which could be treated with antibiotics without surgical intervention and showed almost complete resolution in the recent follow-up. A similar radiological pattern as in our case was observed by Castellanos et al. [10] showing in part cavitation and abscessing. Since *S. aureus* was detected in the initial tracheal aspirate it may have contributed to the clinical and radiological picture as a co-pathogen or even as the culprit.

Reviewing preexisting literature of other systemic *D. pneumosintes* cases indicated the potential to form extensive abscesses. As described in three previous reports there is a high risk of local translocation to form periapical abscess or sinusitis [6,13] and even a frontal brain abscess [14]. The anatomic proximity suggests a continuous venous translocation. Interestingly, *D. pneumosintes* could also be identified in these lastly mentioned cases in anaerobic blood cultures due to a systemic bacteremia. Overall, more than half (six) of the reviewed reports could identify *D. pneumosintes* through blood cultures [4,5,6,13,14,15]. Most recently, a case of 30-year-old previously healthy woman was diagnosed with mediastinal and neck abscess caused by *D. pneumosintes*, after she already received a 2-week course of oral antibiotic for suspected dental abscess [15]. In 4 out of 10 cases, *D. pneumosintes* could be identified through cultivation of material obtained via intervention. Out of these four, *D. pneumosintes* was cultivated in two cases with pneumonia from a bronchoalveolar lavage [10,11] and in the other two cases from pus of a drained brain abscess and liver [14,16]. In total 6 of the 10 reviewed cases needed surgical treatment [6,13,14,15,16].

Six of the reviewed cases show either a sinusal or dental focus as cause for the infection with *D. pneumosintes* [6,13,14,15,16], the other reports don’t further specify the supposed origin of *D. pneumosintes* even though all cases refer to *D. pneumosintes* as a known pathogen of the oropharyngeal cavity causing periodontitis and sinusitis as outlined before.

Of note, blood culture TTP in our case was 34 h. In a recent report, blood culture TTP for clinically relevant anaerobic bacteria was 31.4 ± 23.4 h, indicating at the necessity to consider this fact in the diagnostic and treatment decision process [17]. Moreover, anaerobic infections are usually of polymicrobial nature and frequently require surgical intervention. Therefore, antibiotic regiments should cover aerobic as well as anaerobic antibiotics as we have used in our case (piperacillin/tazobactam, meropenem, ciprofloxacin and sultamicillin) [18].

## 4. Conclusions

In this report we describe the rare case of an adolescent with a severe systemic *D. pneumosintes* infection. Reviewing the preexisting literature, we most likely suspect a hematogenic spread originating from the right-sided sinuses. Since the pathogen is difficult to culture the clinical incidence of systemic infections with *D. pneumosintes* remains unknown, furthermore cohort studies are missing. The reviewed literature indicates that *D. pneumosintes* is potentially causing purulent infections located in the nasopharyngeal cavity but also in different organs. Overall, this work aimed to sensitize health care specialists to consider *D. pneumosintes* infection in patients with periodontal or sinusal infection and to be aware of the possible systemic dissemination leading to severe illnesses including death.

## Figures and Tables

**Figure 1 pathogens-10-00733-f001:**
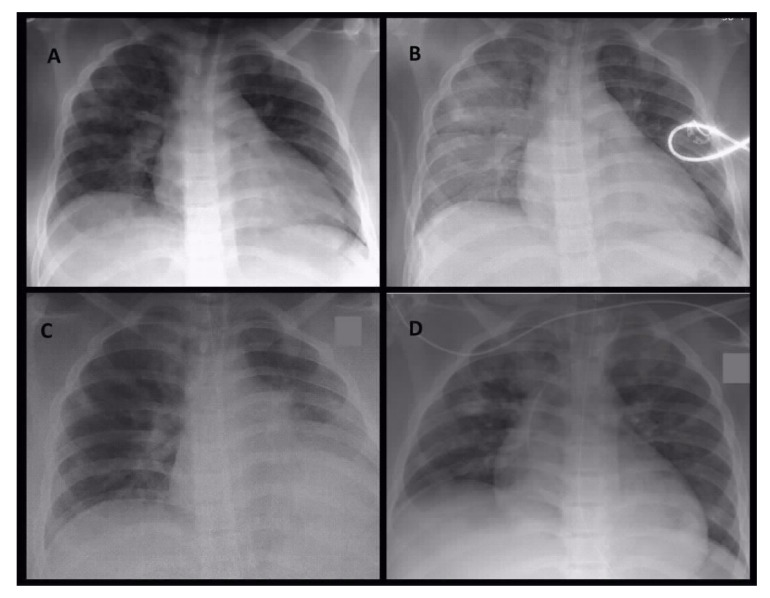
Serial conventional chest X-ray images on day 1 in the morning (**A**), on day 1 at noon (**B**), on day 4 (**C**) and day 5 (**D**). On initial X-ray image (**A**) patchy areas of peribronchovascular infiltrations were detectable. During the same day, ventilation situation became worse with progressive areas of infiltration and retrocardial consolidation (**B)**. These radiological image represent the effect of the infection and not the ventilation. During the stay in intensive care unit ventilation of right side intermediately improved combined with partial atelectasis of left lover lobe (**C**). On day 5 circumscribed areas of consolidations became obvious (**D**), leading to subsequent CT imaging.

**Figure 2 pathogens-10-00733-f002:**
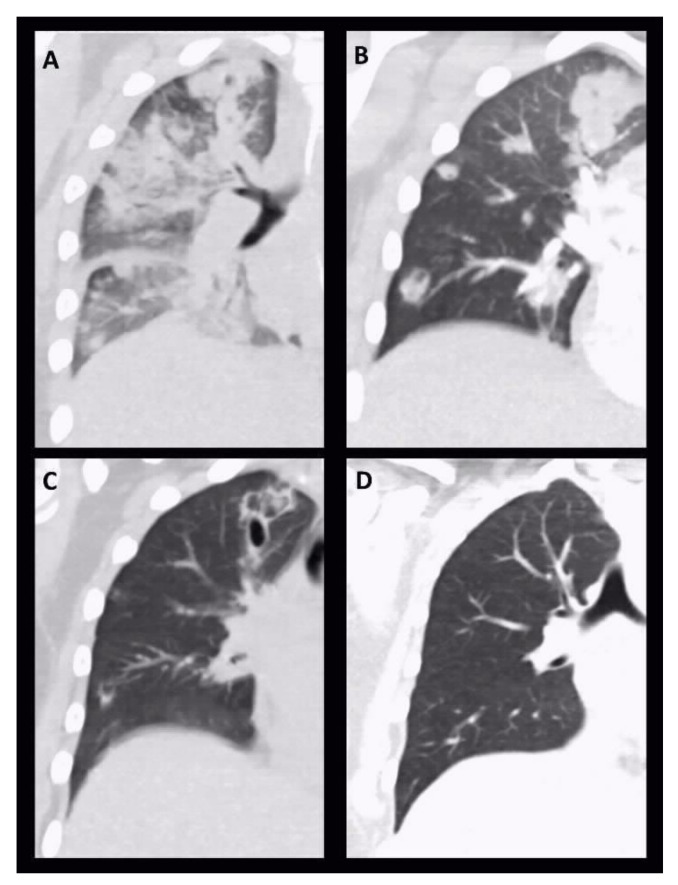
Serial chest CT images: Coronal reconstruction of serial chest CT images on day 1 (**A**), day 6 (**B**), day 17 (**C**) and day 39 (**D**). On initial CT scan periobronchovascular infiltrations with beginning consolidations were detectable. During follow up consolidations became more circumscribed with starting cavitation and abscessing, respectively. In the further course, pneumatoceles initially appeared (**C**), which completely resolved after treatment (**D**).

## Data Availability

Data sharing not applicable.

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
