# Peer review of "Severe Pneumonia and Sepsis Caused by Dialister pneumosintes in an Adolescent"

_pathogens, 2021, doi:10.3390/pathogens10060733_

Round 1
Reviewer 1 Report
Reviewers report
Kaiser et al describe the first case of invasive D. pneumosintes in a young obese girl, convincingly describes the case and goes on to describe other invasive cases of this bacterium. The findings are clearly presented but the manuscript would, in this reviewers opinion, benefit from a few additions in the discussion section and a bit more stringency in the conclusion. I list my comments here in order of scientific priority:
Title and abstract state firstly adolescent, then child… I urge the authors to for reasons of clarity use the same term consequently.
Line 35 Do the authors mean ARDS as an abbreviation of acute or adult respiratory distress syndrome? Why?
Line 42- was the patient checked for viral infection? She was sick since 3 weeks already. CRP levels are usually higher with bacterial infection and lower with viral. Here it is stated 20.2 mg/dl CRP. Also the CRP levels in appendix 1 are quite low. What are the authors comments to this?
Line 49 What tidal volumes were used? If they were appropriate for age and size of the patient this information may be included. It goes towards the infection causing the radiological appearances and not the ventilation.
Line 58 “radiological cause”? How can the authors be sure? Or do the authors mean radiological evidence?
Figure legend 1:
“During same dame (type-o), ventilation situation became worse with progressive areas of infiltration and retrocardial consolidation (B). During the stay in intensive care unit ventilation of right side intermediately improved combined with partial atelectasis of left lover lobe (C). On day 5 circumscribed areas of consolidations became obvious (D),…”
→Seems like an improvement at first. Would the authors care to comment on this? Was such an improvement detected or reported for the other invasive cases that were reviewed?
Line 69-70 “Staphylococcus aureus was further isolated in a tracheal secretion sample as a sign of bacterial co-pathogen superinfection. Repetitive 70 SARS-Cov-2 testing was negative.”
→Why is S aureus not the culprit? The manuscript would benefit, as would the readers, if this was discussed in the appropriate section.
How much relates to patient susceptibility and how much relates to virulence of the bacteria? With regard to this, what role does her obesity play? Please elaborate if obese is to be stated in the title.
Appendix 1 is a great overview.
The authors should be aware and include this newly published paper
Am J Case Rep. 2021 Mar 27;22:e930559. doi: 10.12659/AJCR.930559.
A Case of Dialister pneumosintes Bacteremia-Associated Neck and Mediastinal Abscess
Sonia Mannan 1 , Tahir Ahmad 1 , Asma Naeem 2 , Vinod Patel 3
Affiliations
- PMID: 33772571 DOI: 10.12659/AJCR.930559
A few minor flaws of English language or suspected misspellings:
- Line 32 2 errors: been is repeated, the wording is strange- frequently is placed in the wrong position in the sentence.
- Line 52 along with (with is missing)
- Line 56 consider emerging instead of beginning- or beginning OF
- Figure legend 1 “During same dame (type-o), ventilation situation became worse with progressive areas of infiltration and retrocardial consolidation (B). During the stay in intensive care unit ventilation of right side intermediately improved combined with partial atelectasis of left lover lobe (C). On day 5 circumscribed areas of consolidations became obvious (D),…”
And out of curiosity:
- Line 116 pneumosintes was formerly named Bacteroides pneumosintes. Why was not the former name included in the literature searches in Pubmed?
With my kind regards,
Author Response
Kaiser et al describe the first case of invasive D. pneumosintes in a young obese girl, convincingly describes the case and goes on to describe other invasive cases of this bacterium. The findings are clearly presented but the manuscript would, in this reviewer`s opinion, benefit from a few additions in the discussion section and a bit more stringency in the conclusion. I list my comments here in order of scientific priority:
Title and abstract state firstly adolescent, then child… I urge the authors to for reasons of clarity use the same term consequently.
Response: We thank the reviewer for the comment and changed the text accordingly.
Line 35 Do the authors mean ARDS as an abbreviation of acute or adult respiratory distress syndrome? Why?
Response: The child had clinically an acute respiratory distress syndrome. We explained the abbreviation in the text, now.
Line 42- was the patient checked for viral infection? She was sick since 3 weeks already. CRP levels are usually higher with bacterial infection and lower with viral. Here it is stated 20.2 mg/dl CRP. Also the CRP levels in appendix 1 are quite low. What are the authors comments to this?
Response: A CRP of 20.2 mg/dl is high (202 mg/l). Repetitive SARS-Cov-2 testing was negative. Other Viral pathogens were not looked after at this stage.
Line 49 What tidal volumes were used? If they were appropriate for age and size of the patient this information may be included. It goes towards the infection causing the radiological appearances and not the ventilation.
Response: The tidal volumes were adapted for ARDS treatment and radiological images represented the effect of the infection and not the ventilation. This information is provided in the text, now.
Line 58 “radiological cause”? How can the authors be sure? Or do the authors mean radiological evidence?
Response: We thank the reviewer for the comment and changed the text accordingly.
Figure legend 1:
“During same dame (type-o), ventilation situation became worse with progressive areas of infiltration and retrocardial consolidation (B). During the stay in intensive care unit ventilation of right side intermediately improved combined with partial atelectasis of left lover lobe (C). On day 5 circumscribed areas of consolidations became obvious (D),…”
→Seems like an improvement at first. Would the authors care to comment on this? Was such an improvement detected or reported for the other invasive cases that were reviewed?
Response: The typo in text was changed accordingly. Unfortunately, the lung infection cases in the literature were not described in detail.
Line 69-70 “Staphylococcus aureus was further isolated in a tracheal secretion sample as a sign of bacterial co-pathogen superinfection. Repetitive 70 SARS-Cov-2 testing was negative.”
→Why is S aureus not the culprit? The manuscript would benefit, as would the readers, if this was discussed in the appropriate section.
Response: We thank the reviewer for the comment and discussed this aspect in the discussion section of the manuscript now.
How much relates to patient susceptibility and how much relates to virulence of the bacteria? With regard to this, what role does her obesity play? Please elaborate if obese is to be stated in the title.
Response: Thanks for mentioning the interesting aspects, that are discussed in the discussion section of the manuscript, now.
Appendix 1 is a great overview.
The authors should be aware and include this newly published paper
Am J Case Rep. 2021 Mar 27;22:e930559. doi: 10.12659/AJCR.930559.
A Case of Dialister pneumosintes Bacteremia-Associated Neck and Mediastinal Abscess
Sonia Mannan 1 , Tahir Ahmad 1 , Asma Naeem 2 , Vinod Patel 3
Affiliations
- PMID: 33772571 DOI: 10.12659/AJCR.930559
Response: We thank the reviewer for the comment and the new citation, that we have included, now.
A few minor flaws of English language or suspected misspellings:
- Line 32 2 errors: been is repeated, the wording is strange- frequently is placed in the wrong position in the sentence.
- Line 52 along with (with is missing)
- Line 56 consider emerging instead of beginning- or beginning OF
- Figure legend 1 “During same dame (type-o), ventilation situation became worse with progressive areas of infiltration and retrocardial consolidation (B). During the stay in intensive care unit ventilation of right side intermediately improved combined with partial atelectasis of left lover lobe (C). On day 5 circumscribed areas of consolidations became obvious (D),…”
Response: We thank the reviewer for the comment and corrected all spelling mistakes.
And out of curiosity:
- Line 116 D. pneumosintes was formerly named Bacteroides pneumosintes. Why was not the former name included in the literature searches in Pubmed?
Response: For clarity reasons we have concentrated on the new term and not clear-cut cases we described in the old literature.
Reviewer 2 Report
The authors should address the following concerns:
General:
the authors should check the MS for punctuation issues, Oxford-comma etc.
antibiotic names, and laboratory parameters (e.g. procalcitonin) should not be capitalized
Introduction:
can you really consider Dialister as „non-fermentative” (similar to when comparing to the distinction between fermenters and non-fermenters)? If so, please provide a reference
„non-fermentative”
L31-32: please rephrase this sentence
„up to”
„systemic infection”
„13-year-old”
„Figure 1”
„Figure 2”
L61: one
time-to-positivity (TTP)
Gram-staining
ceftriaxone
L128-133: please include the following reference on the relevance of anaerobic bacteria in bacteremia and the relevance of TTP in determining the pathogenic role of anaerobic bacteria in sepsis:
https://pubmed.ncbi.nlm.nih.gov/32247001/
Please discuss the possible therapeutic options in anaerobic infections considering the following references:
https://pubmed.ncbi.nlm.nih.gov/26620376/
Author Response
The authors should address the following concerns:
General:
the authors should check the MS for punctuation issues, Oxford-comma etc.
antibiotic names, and laboratory parameters (e.g. procalcitonin) should not be capitalized
Response: We thank the reviewer for the comment and corrected accordingly.
Introduction:
can you really consider Dialister as „non-fermentative” (similar to when comparing to the distinction between fermenters and non-fermenters)? If so, please provide a reference
„non-fermentative”
Response: The reference Doan et al. 2000 described Dialister as “non-fermentative”.
L31-32: please rephrase this sentence
„up to”
„systemic infection”
„13-year-old”
„Figure 1”
„Figure 2”
Response: We thank the reviewer for the comment and corrected accordingly.
L61: one
time-to-positivity (TTP)
Gram-staining
Ceftriaxone
Response: We thank the reviewer for the comment and corrected accordingly.
L128-133: please include the following reference on the relevance of anaerobic bacteria in bacteremia and the relevance of TTP in determining the pathogenic role of anaerobic bacteria in sepsis:
https://pubmed.ncbi.nlm.nih.gov/32247001/
Response: We thank the reviewer for the reference and cited it accordingly in the discussion section of the manuscript.
Please discuss the possible therapeutic options in anaerobic infections considering the following references:
https://pubmed.ncbi.nlm.nih.gov/26620376/
Response: We thank the reviewer for the reference and cited it accordingly in the discussion section of the manuscript.